# Proteomic and Metabolomic Analysis of the *Quercus ilex–Phytophthora cinnamomi* Pathosystem Reveals a Population-Specific Response, Independent of Co-Occurrence of Drought

**DOI:** 10.3390/biom14020160

**Published:** 2024-01-29

**Authors:** Tamara Hernández-Lao, Marta Tienda-Parrilla, Mónica Labella-Ortega, Victor M. Guerrero-Sánchez, María-Dolores Rey, Jesús V. Jorrín-Novo, María Ángeles Castillejo-Sánchez

**Affiliations:** Agroforestry and Plant Biochemistry, Proteomics and Systems Biology, Department of Biochemistry and Molecular Biology, University of Cordoba, UCO-CeiA3, 14014 Cordoba, Spain; b42helat@uco.es (T.H.-L.); b72tipam@uco.es (M.T.-P.); b62laorm@uco.es (M.L.-O.); b12gusav@uco.es (V.M.G.-S.); b52resam@uco.es (M.-D.R.)

**Keywords:** *Quercus ilex*, decline syndrome, climate change, markers of resilience, proteomics, metabolomics

## Abstract

Holm oak (*Quercus ilex*) is considered to be one of the major structural elements of Mediterranean forests and the agrosilvopastoral Spanish “dehesa”, making it an outstanding example of ecological and socioeconomic sustainability in forest ecosystems. The exotic *Phytophthora cinnamomi* is one of the most aggressive pathogens of woody species and, together with drought, is considered to be one of the main drivers of holm oak decline. The effect of and response to *P. cinnamomi* inoculation were studied in the offspring of mother trees from two Andalusian populations, Cordoba and Huelva. At the two locations, acorns collected from both symptomatic (damaged) and asymptomatic (apparently healthy) trees were sampled. Damage symptoms, mortality, and chlorophyll fluorescence were evaluated in seedlings inoculated under humid and drought conditions. The effect and response depended on the population and were more apparent in Huelva than in Cordoba. An integrated proteomic and metabolomic analysis revealed the involvement of different metabolic pathways in response to the pathogen in both populations, including amino acid metabolism pathways in Huelva, and terpenoid and flavonoid biosynthesis in Cordoba. However, no differential response was observed between seedlings inoculated under humid and drought conditions. A protective mechanism of the photosynthetic apparatus was activated in response to defective photosynthetic activity in inoculated plants, which seemed to be more efficient in the Cordoba population. In addition, enzymes and metabolites of the phenylpropanoid and flavonoid biosynthesis pathways may have conferred higher resistance in the Cordoba population. Some enzymes are proposed as markers of resilience, among which glyoxalase I, glutathione reductase, thioredoxin reductase, and cinnamyl alcohol dehydrogenase are candidates.

## 1. Introduction

Invasion by exotic pests and pathogens, together with global change drivers such as climate change, is causing forest decline worldwide, and this is increasing at an unprecedented rate [1,2]. The Mediterranean Basin is considered to be among the regions most susceptible to global change, and climate models predict a 2–5 °C increase in temperature and a 30% reduction in annual rainfall by the end of the 21st century in this region [3,4]. In addition, exotic pathogens, particularly those belonging to the genus *Phytophthora*, are causing devastating epidemics in this area. However, it is uncertain how the simultaneous and interactive effects of exotic pathogens and a drier climate will affect the regeneration dynamics of Mediterranean forests [5,6]. Most *Phytophthora* species are polyphagous with a range of host plants. Many *Phytophthora* species, such as *Phytophthora infestans, P. cinnamomi*, and *P. ramorum*, are destructive plant pathogens that cause severe crop losses and tree declines worldwide [2,5,6,7]. *Phytophthora cinnamomi* is one of the most aggressive pathogens of woody species [5,8]; it causes rot root and destroys the fine roots of the hosts, impeding nutrient and water uptake, with the consequent defoliation, loss of vigor, and even death of infected trees. However, the presence of multiple *Pytophthora* species (*P. cinnamomi, P. quercina,* and *P. gonapodyides*) in the same place can result in a faster decline of holm oak forests [9]. *Phytophthora cinnamomi* has been ranked in the top 10 oomycete plant pathogens based on scientific and economic importance [7]. This pathogen has been detected in 15 biodiversity hotspots, including the Mediterranean Basin [10]. 

Evergreen oaks such as holm oak (*Quercus ilex*) and cork oak (*Quercus suber*) are considered to be the major structural elements of Mediterranean forests and the agrosilvopastoral Spanish “dehesa” or Portuguese “montado”, making them outstanding examples of the ecological and socioeconomic sustainability of forest ecosystems [11,12]. *Quercus ilex* dominates the Mediterranean Basin, covering more than 6 million ha [13,14]. The heterozygosity and allelic richness of this species may explain its wide ecological reach and ecophysiological adaptability to extreme conditions [15,16,17]. However, the species is threatened by several factors associated with anthropogenic activity, such as the overexploitation and poor regeneration of livestock, and the combination of biotic and abiotic stresses, such as drought and high temperatures, together with the presence of invasive pathogens, causing so-called decline syndrome [18,19,20]. The soil-borne pathogen *P. cinnamomi* is considered to be one of the main drivers of holm oak decline [21,22,23,24]. It is predicted that global climate change will increase the incidence of *P. cinnamomi* in forest and crop species, promoting the extinction of endemic species in vulnerable areas [5,25]. The progressive loss of resilience revealed for holm oak might have to do with the timing of drought events, which, together with increased infestation by *P. cinnamomi*, will likely undermine the holm oak domain [26]. In declined areas, the presence of healthy (asymptomatic) trees close to diseased (symptomatic) ones has been observed [27,28]. Healthy trees can be considered escape individuals or elite genotypes based on the resiliency of their phenotype to adverse conditions. The contribution of genetic and epigenetic factors to intra- and inter-population *Q. ilex* phenotypic variability remains unknown, and to date, there have not been many molecular studies addressing this issue.

The main changes in plant status caused by *P. cinnamomi* infection have been related to physiological responses, such as stomatal closure, photosynthesis rate, and water imbalance [29,30,31]. As reported by Sghaier-Hammami et al. [29], the response of *Q. ilex* seedlings to *P. cinnamomi* resembles the drought stress response, with differences between provenances; a general decrease in proteins related to photosynthesis and an increase in starch biosynthesis were observed. A differential response to *P. cinnamomi* was observed between *Q. suber* and *Q. variabilis* upon using a combined proteomic and metabolomic approach [32]. The biggest changes in the leaf proteome included increased abundance of peroxidases, superoxide dismutases, and glutathione S-transferases in *Q. variabilis*, which probably contribute to decreased susceptibility to *P. cinnamomi*.

The effect of *P. cinnamomi* in combination with water deprivation on *Q. ilex* seedlings has also been studied at the phenotypic, physiological, biochemical, and molecular levels [30,33,34]. Previous studies demonstrated that drought increases susceptibility to *P. cinnamomi* in *Q. ilex* seedlings [9,30,33]. Ruiz-Gomez et al. [33] reported that the root mass fraction of *Q. ilex* seedlings was significantly reduced by *P. cinnamomi* infection, altering the growth pattern. However, the plants could not recover from the physiological effects of infection when the root rot coincided with water stress.

The combination of drought and *P. cinnamomi* was demonstrated to have a synergistic effect on *Q. ilex* seedlings from three contrasting Andalusian (Spanish) populations [34]. The authors reported that the easternmost population was the most susceptible to the combined stress, with a high proportion of stress-related proteins identified using a proteomic approach [34]. The proteins aldehyde dehydrogenase, glucose-6-phosphate isomerase, 50S ribosomal protein L5, and α-1,4-glucan-protein synthase were proposed as putative markers of resilience to the decline syndrome.

In order to elucidate the metabolic pathways involved in resistance against *P. cinnamomi,* the effect of inoculation, in well-watered and non-watered holm oak seedlings, was studied from morphophysiological and molecular (proteomics and metabolomics) perspectives. For this purpose, the offspring of mother trees, both symptomatic (with visual damage symptoms) and asymptomatic (without visual damage symptoms), from two Andalusian populations, Cordoba and Huelva, were surveyed. The results were compared to those obtained previously in similar *Quercus* studies. From the present work, a list of candidate genes is proposed as putative markers of resistance against *P. cinnamomi* to be used in breeding programs for resilience. 

## 2. Materials and Methods

### 2.1. Plant Material

*Quercus ilex* mother trees from two populations, Huelva (Hu; Santa Olalla del Cala and Aroche) and Córdoba (Co; Hinojosa del Duque and El Viso), were surveyed (Appendix A). In these two areas, the presence or absence of *P. cinnamomi* has been tested through soil baiting and morphological characterization [24]. Acorns from symptomatic (with visual damage symptoms) and asymptomatic (without visual damage symptoms) individuals were collected in autumn 2020, their surface was cleaned with 2.5% bleach solution, and floating acorns were discarded. Then, acorns were extensively washed again with tap water and blotted dry. To obtain a uniform germination rate, the pericarp was removed and the distal half of the cotyledon was cut out [35]. Germination was carried out in darkness at 25 °C in plastic trays (28 cm × 7 cm × 18 cm, about 60 acorns per tray) on a bed of perlite flooded with tap water and covered with a sheet of moistened filter paper. Seven days after imbibition, germinated seeds (with roots reaching 5–6 cm in length) were sown individually in plastic pots (3 L) using perlite as the substrate (Gramoflor GmbH & Co, Vechta, Germany). Sown seedlings were transferred to a climate-controlled greenhouse for growth under the following conditions: 25 ± 10 °C and 60 ± 10% air humidity, a 11/13 h photoperiod, 900 μmol (photons) m^−2^ s^−1^ light. All pots were watered with 200 mL tap water every 2 days and once a week with Hoagland solution [36] until plants reached 6 months old. 

### 2.2. Experimental Design

Six-month-old seedlings were subjected to four treatments: (1) control (C; irrigation to field capacity, non-inoculated with *P. cinnamomi*), (2) drought (D; no irrigation, non-inoculated with *P. cinnamomi*), (3) inoculation (I; irrigation, inoculated with *P. cinnamomi* and irrigated), and (4) combined stress (D × I, inoculated with *P. cinnamomi* and non-irrigated) (Appendix A). 

Twelve seedlings with similar morphological characteristics in terms of height and number of leaves were selected per population (Hu and Co), adult tree (with and without damage), and treatment (C, D, I, and D × I) to perform the experiment. The *P. cinnamomi* strain (P90) was isolated from *Q. ilex* roots in Puebla de Guzmán (Huelva, Spain) and provided by Prof. Rafael M. Navarro Cerrillo [12,24,33,37]. The treatments were performed as described by Ruiz-Gómez et al. [33] and San-Eufrasio et al. [34]. Briefly, inoculation with *P. cinnamomi* was performed via root immersion for 10 min in carrot–agar (CA) liquid medium at a concentration of 43 chlamydospores/μL [33]. Roots of control seedlings were also immersed in sterile CA liquid without *P. cinnamomi*. After inoculation, seedlings were transplanted into pots with fresh perlite and flooded for 48 h. Then, excess water was removed before drought treatment. Drought was imposed by withholding water for 28 days [38]. The non-drought treatments included watering with 200 mL tap water every 2 days and once a week with Hoagland solution [36]. The presence of *P. cinnamomi* in the inoculated plants was confirmed on day 12 of the experiment by using root segments (<2 mm thick, ~1 cm long) placed in Petri dishes of selective medium (PARPBH), as described Ruiz-Gomez et al. [33] and San-Eufrasio et al. [34].

### 2.3. Morphological and Physiological Measurements

Leaf chlorosis, wilting, and senescence were visually observed throughout the experiment. Damage symptoms were quantified every three days according to an empirical scale described by San-Eufrasio et al. [38]: 0 = no leaves showing symptoms; 1 = one or two leaves showing slight drought symptoms (necrosis along edges and/or veins, changes in color of foliage (light green, yellow, or brown), and/or irregular spot changes); 2 = most leaves showing slight drought symptoms but one or two leaves still not showing symptoms; 3 = all leaves showing drought symptoms but they are not severe; 4 = all leaves showing severe drought symptoms (leaves with a totally dry yellow aspect); and 5 = whole seedling showing wilting and/or falling of leaves. 

Quantum yield of photosystem II (Qy) was measured with a FluorPen FP100 portable leaf fluorimeter from Photon Systems Instruments (Drasiv, Czech Republic) every 3 days throughout the experiment. Three seedlings per treatment per population were selected to perform Qy measurements, always using the same three youngest fully expanded leaves of each plant, in the early morning when the leaves were adapted to nighttime darkness, according to San-Eufrasio et al. [38]. 

Seedlings with all leaves exhibiting severe symptoms (classified as 4–5 on the damage scale) and Qy values near 0 were considered to be dead.

### 2.4. Protein Extraction and GeLC-MSMS Analysis

Proteomic analysis was performed on leaf tissue from the offspring of asymptomatic mother trees. Leaves were sampled when chlorophyll fluorescence decreased to values in the 0–75% range with respect to the control, with a mean value of 35% (which occurred on day 12), in plants subjected to stress. Healthy leaves from three biological replicates per treatment and population were collected, washed with distilled water, dried with paper, and frozen in liquid nitrogen. Proteins were extracted from leaf tissue (300 mg) using the TCA/acetone-phenol protocol [39] and solubilized in a solution containing 7 M urea, 2 M thiourea, 4% (*w*/*v*) 3-[(3-cholamidopropyl)dimethylammonio]-1-propanesulfonate (CHAPS), 0.5% (*w*/*v*) Triton X-100, and 100 mM DTT. Protein extracts were then quantified via the Bradford method using bovine serum albumin (BSA) as the standard [40].

Extracted proteins (80 μg) were subjected to SDS-PAGE on 12% polyacrylamide gel (8.6 × 6.7 cm^2^) in a Mini-PROTEAN Cell (Bio-Rad, Hercules, CA, USA). The gel was run at 80 V and stopped when the bromophenol blue had advanced 0.5 cm into the resolving gel. The gel was stained with Coomassie brilliant blue R-250, and the resulting unique band was cut from the gel with a scalpel and transferred to individual 1.5 mL tubes for trypsin digestion and mass spectrometry analysis [41]. MS analysis was conducted at the Proteomics Facility for Research Support Central Service (SCAI) of the University of Cordoba (Spain) using a Thermo Scientific Orbitrap Fusion Tribrid mass spectrometer operated in positive ion mode. The specific settings used in the LC–MS/MS analysis are described elsewhere [42]. 

### 2.5. Protein Identification and Quantification

The raw data were processed using Proteome Discoverer v.2.3 software from Thermo Scientific. The SEQUEST engine was used to search MS2 spectra against the FASTA Quercus_database obtained from the translation of the *Q. ilex* transcriptome [43,44]. A maximum of two missed cleavages of trypsin was used in all searches. Precursor mass tolerance was set at 10 ppm, fragment ion mass tolerance at 0.1 Da, and charge state at +2 or greater. Peptides were filtered to 1% of the false discovery rate (FDR) and classified into proteins according to the law of parsimony.

The parameters for protein confidence identification were set to at least 2 matching peptides, a minimum score of 2, and sequence coverage mostly greater than 10%. Proteins were quantified in relative terms from the peak areas for precursor ions (average of the three strongest peptide ion signals) [45]. Raw data were deposited in the ProteomeXchange Consortium via PRIDE [46], under the identifier PXD046909. Protein values were normalized taking into account the relative protein abundance between samples using the sum of the peak area values for each sample (Appendix A). Consistent values for at least three biological replicates of the same condition were considered for further statistical analysis. Then, proteins were categorized by function from their FASTA sequences using the online tool Mercator v.3.6 (MapMan; accessed on 10 July 2023) [47].

### 2.6. Metabolite Extraction and LC-MS Analysis

The metabolomic analysis was performed at the same sampling time as the proteomics analysis (day 12), with seedlings from asymptomatic Co and Hu mother trees subjected to *P. cinnamomi* inoculation without irrigation. Metabolites were extracted from freeze-dried leaf powder. Briefly, a buffer containing 1200 μL of cold ethanol/water (80:20) was added to 30 mg of leaf powder, and tissue disruption was driven by maceration with a pistil, followed by vortexing (10 s) and sonication (ultrasonic bath, 40 kHZ for 10 min). After centrifugation (16,000× *g*, 4 °C, 6 min), the supernatant was vacuum dried at 30 °C (Eppendorf 5301 Vacufuge Concentrator Plus, Eppendorf, Leicestershire, UK) and reconstituted in 0.5 mL of 50% methanol containing 0.1% of formic acid, centrifuged (20,000× *g*, 10 min), and filtered through 0.22 μm PTPE membranes (Thermo Fisher Scientific, Courtaboeuf, France). The filtrate was collected in 1.5 mL LC/MS certified sample vials. 

All analyses were performed using a liquid chromatography system consisting of a binary UHPLC Dionex Ultimate 3000 RS connected to a Q Exactive Plus Hybrid Quadrupole–Orbitrap mass spectrometer (Thermo Fisher Scientific, Bremen, Germany), which was equipped with a heated-electrospray ionization probe (HESI-II). Chromatographic separations were performed using an Acquity UPLC BEH C18 column (2.1 × 100 mm, 1.7 µm) (Waters). The column was maintained at 40 °C and eluted under the following conditions: 5% B for 1 min, linear gradient from 5% to 100% in solvent B for 9 min, isocratic at 100% B for 2 min, return to initial conditions, 5% B for 3 min. A flow rate of 0.5 mL/min was used. Eluent A was 0.1% formic acid in water and eluent B was 0.1% formic acid in methanol. Injection volume was 5 µL.

MS detection was performed with the Q Exactive Orbitrap mass spectrometer operating in positive and negative polarities. HESI source parameters in positive mode were as follows: spray voltage of 3.5 kV; S-lens RF level of 50; capillary temperature of 320 °C; sheath and auxiliary gas flow of 60 and 25, respectively (arbitrary units); and probe heater temperature of 400 °C. For negative ion mode, all parameters remained the same, except that the spray voltage was set to −3.0 kV. Xcalibur v.4.3 software was used for instrument control and data acquisition. Full Scan MS mode was used at a resolution of 70,000 (full width half maximum (FWHM) at *m*/*z* 200) and data-dependent acquisition (DDA) MS2 mode was used at resolutions of 70,000 and 17,500 (FWHM at *m*/*z* 200) for Full Scan and Product Ion Scan, respectively, fragmenting the five most abundant precursor ions per MS scan (Top5). Full Scan MS and DDA-MS2 were used in positive and negative modes, and the mass range used for both experiments was 70–1050 *m*/*z*.

### 2.7. Metabolite Identification and Data Processing

Data were exported for analysis by Compound Discoverer v3.2 software (Thermo Fisher Scientific, Bremen, Germany), in which MS data treatment, alignment, peak selection, deconvolution, normalization, and annotation were carried out. Alignment was performed using a retention time with a maximum shift of 0.3 min and a mass tolerance of 3 ppm. Metabolites were quantified in relative terms considering the area of the corresponding chromatographic peaks of the MS1 precursor ion. Neutral masses obtained in positive and negative modes were evaluated to avoid duplicates (neutral mass in different modes and similar retention time) while retaining the most intense peaks. Metabolite values were normalized taking into account the relative abundance between samples, using the sum of the peak area for each sample. Values consistently present in at least three biological replicates of the same condition were considered for further statistical analysis. A table was generated that lists the abundance of metabolites in the samples (Appendix A). Metabolite annotations were developed using DDA-MS2 similarity and the formula or exact mass searched in mzCloud and ChemSpider, respectively. Finally, annotations of the ions of interest were manually reviewed and classified into metabolic pathways (KEGG database; accessed November 2023) [48] and chemical families (ClassyFire; accessed on 15 November 2023) [49]. Raw data were deposited in the NIH Common Fund’s National Metabolomics Data Repository (NMDR) on the Metabolomics Workbench website (accessed on 12 December 2023) [50]. The data can be accessed directly via DataTrack ID: 4480.

### 2.8. Statistical Analysis

The area under the Qy curve was calculated [38] and data were analyzed using the Kruskal–Wallis test (nonparametric one-way analysis of variance). Means were separated using Dunn’s test with a probability level of 0.05. Statistical analyses were performed using STATISTIX 10.0 software (Analytical Software, Tallahassee, FL, USA).

Proteomics and metabolomics statistical analyses were performed using three biological replicates per treatment and population. Multivariate analysis (principal component analysis (PCA)) was performed for both proteomics and metabolomics using the FactoMineR package in R v4.2.1 [51]. The non-parametric Kruskal–Wallis test was applied to determine statistically variable proteins and metabolites between treatments using the stats package in R [52]. Proteins and metabolites showing significant differences (*p* ≤ 0.05) that were up-accumulated under the different stress conditions were used for downstream analysis. Venn diagrams were generated using *ggvenn* function from R package ggven [53] to visualize unique and common proteins and metabolites between populations with the same treatment and between treatments for each population. 

### 2.9. Integrative Analysis of Proteomics and Metabolomics Data

Pathway enrichment analysis was carried out to investigate the pathways significantly activated under *P. cinnamomi* inoculation. For this, the significantly up-accumulated proteins and metabolites specific to and common between populations and treatments were manually mapped to their respective pathways using the Kyoto Encyclopedia of Genes and Genomes (KEGG) database (accessed on 14 November 2023) [48]. For this, the EC number for enzymes and KEGG ID for metabolites were used. 

Multi-omics factor analysis (MOFA) is a computational method for discovering the principal sources of variation in multi-omics datasets. This analysis infers a set of hidden or latent factors that capture the biological and technical sources of variability [54]. Proteomic and metabolomic data were processed and integrated using the MOFA2 package (version 1.10.0) in R (version 4.3.1). To prepare for model training, two sets of the combined stress-regulatory layers were used: proteins and metabolites. Model training was carried out using the following options: maxiter = 1000, convergence_mode = slow.

## 3. Results and Discussion

The effects of and responses to *P. cinnamomi* were analyzed in offspring from holm oak mother trees in the Hu and Co populations (Appendix A). They corresponded to declined and non-declined areas from the two locations, with the presence of the pathogen in the soil from the declined areas confirmed. In both locations, acorns were collected from trees that both showed and did not show damage symptoms (Appendix A). Damage symptoms included dark brown branches and around 40% defoliation. Symptomatic damaged trees also showed lower photosynthetic efficiency than asymptomatic ones (leaf fluorescence values of 0.8 and below 0.5 for damaged and non-damaged trees, respectively). 

According to the experimental design, we evaluated and compared the response to *P. cinnamomi* in offspring based on the individual (genotype), provenance, coexistence (or not) with the pathogen, and health status (with or without damage symptoms). A similar experiment was conducted by Vivas et al. [55] using different provenances. They observed a higher probability of survival for seedlings from *P. cinnamomi*-infected trees than seedlings from non-infected trees, particularly seedlings with reduced growth. As *P. cinnamomi* and drought are the main causes of decline syndrome [33,34], as well as unpredictable consequences in climate change scenarios [1], the inoculation experiment was performed with well-watered and non-watered seedlings (corresponding to a soil matrix potential of 0 and −33 kPa at the end of the experiment for control watered and non-watered seedlings, respectively). During the 16-day experiment, damage symptoms, mortality, and the quantum yield of photosystem II (Qy) were evaluated. Proteomic and metabolomic analyses were performed with the offspring from the asymptomatic non-damaged mother trees of both populations on day 12, when leaf chlorophyll fluorescence decreased to an average of 35% of the initial value in plants subjected to the treatments relative to the control plants. The inoculation of plants under both water regimes resulted in successful infection of the root system in seedlings from both symptomatic and asymptomatic offspring, as revealed by visual and microscopic observation of *P. cinnamomi* mycelium in isolated root fragments sown in selective PARPBH medium (Appendix A) [56].

### 3.1. Seedling Damage Symptoms, Mortality, and Leaf Fluorescence

Leaf damage symptoms (chlorosis, necrosis, and wilting) and mortality depended on the individual and the treatment (Appendix A). All offspring from Hu mother trees, in all inoculated treatments, showed 100% mortality on day 16 (Figure 1). The mortality rate of offspring from asymptomatic Co trees was considerably lower than that of symptomatic trees. The rates for irrigated and non-irrigated inoculated treatments were 22% and 33% for the former, and 50% and 90% for the latter.

In general, mortality was higher for the Hu than the Co population, suggesting that the latter has higher resistance to *P. cinnamomi*. This difference was clearly observed when evaluating mortality at the midpoint of the experiment (Figure 1a,b), when the following comparative levels of resistance were found: Hu_symptomatic < Co_symptomatic < Hu_asymptomatic < Co_asymptomatic. This confirms the higher resistance of asymptomatic than symptomatic genotypes in both provenances. This finding differs from that reported by Vivas et al. [55]; in that study, resistance was higher in the offspring of infected than non-infected mother trees. In general, withholding water had a small effect at 12 days, when leaves were sampled for proteomic analysis (mortality rate lower than 30% for Hu population and around 40% for Co) (Figure 1a,b). In a previously published similar experiment, the drought effect was observed at later times [57]. This is because, at least for the two Hu individuals, the main effect observed was due to the inoculation. For the Co population, withholding water increased the rate of mortality of the offspring of symptomatic individuals, but not those of asymptomatic trees (Figure 1a). These data confirm the complexity of the plant–pathogen interaction and the resulting phenotype, determined by genetics (maternal inheritance), as can occur in the Co population; epigenetics [58,59]; or environmental factors (presence or lack of the pathogen in the soil, among other factors). As the density of the inoculum in the soil of the mother trees and the father of the offspring are unknown, corresponding to an allogamous species, it is only possible to speculate on resistance and its inheritance. 

Leaf fluorescence is an estimating parameter for the quantum yield of photosystem II (Qy) and has been used as a parameter for the response to stress in plants [60,61,62]. Its value in the control plants was around 0.7 (Figure 1c,d). Inoculation caused a decrease in fluorescence values, and this effect was more apparent in Hu than Co. A significant decrease was observed due to the effect of withholding water and inoculating in the symptomatic Co offspring (Figure 1c), while there were no differences between symptomatic and asymptomatic Hu seedlings (Figure 1d). These data confirm the previous hypothesis of higher resistance to oomycetes in the Co population than the Hu population and between individuals within the Co population. 

### 3.2. Proteomic Analysis

Proteomic analysis was performed with the leaf tissue of seedlings from asymptomatic mother trees of both populations under the control (well watered and non-inoculated) and *P. cinnamomi*-inoculated (watered and non-watered) treatments. The leaves were sampled on day 12, when chlorophyll fluorescence on the leaves of the inoculated plants had decreased on average by 35% with respect to the control.

The average protein extraction yield from the leaves was 2.7 mg/g fresh weight, similar to that obtained from other holm oak provenances [63], with no significant differences between samples. For the shotgun analysis, 80 µg of protein from each sample was analyzed, allowing the identification of 3446 total protein species (2735 for Co and 3019 for Hu). The raw data were filtered for confident identification and quantification. Proteins with two or more peptides and a score higher than 2 that were consistently present in all three replicates in at least one condition were selected. This resulted in 2565 total proteins being confidently identified, 1590 in Co and 1444 in Hu individuals, with 1334 common to both (Table 1, Appendix A). These data confirm the great variability commonly found in *Q. ilex*, with the protein profile depending on the genotype [64]. Two-dimensional PCA explained 40 and 43% of the total variability in Co and Hu, respectively (Figure 2a,b). PC1 (27.2%) correlated to some extent with the control, drought, and inoculation treatments in Co seedlings; on the contrary, it did not correlate with treatments in the Hu population. 

The dataset of confident proteins was then subjected to the Kruskal–Wallis statistical test, which revealed 600 and 525 proteins for Co and Hu, respectively, with 350 common to both and significant differences between treatments (D, I, and D × I) compared to the control (Table 1). Variable proteins were classified into 14 main functional groups (Appendix A): carbohydrate, amino acid, lipid, secondary, and nucleotide metabolism; photosynthesis; stress; redox; transcription/translation; signaling; transport, folding, sorting, and degradation; development; and cellular processes. The variable up-accumulated proteins for Co and Hu, respectively, were 33 and 20 in response to drought, 49 and 61 in response to *P. cinnamomi* with watering, and 43 and 65 in response to *P. cinnamomi* without watering (Table 1, Appendix A). Regarding down-accumulated proteins, the largest proportions were photosynthesis and folding, sorting, and degradation, specifically in the Hu population (Appendix A).

Among the variable proteins, as shown in Table 2 and Figure 3, 182 were up-accumulated in response to stress compared to the control plants. The pattern of these proteins depended on the genotype and treatment. There were more treatment-specific proteins for Hu than Co (69 vs. 59) (Figure 3a,b, Table 2). Regarding common proteins between treatments, 31 were found for Co and 37 for Hu, and 14 were common to both (Figure 3c, Table 2). No common proteins that were represented more in the combined stress than the inoculated groups were found between the two populations. Among the 14 common proteins, the most represented group was carbohydrate metabolism, with 7 proteins: chloroplastic fructose-1,6-bisphosphatase, phosphoglycerate mutase, beta-xylosidase/alpha-L-arabinofuranosidase 2, beta-galactosidase, pyruvate dehydrogenase E1 component subunit alpha mitochondrial, and phosphoenolpyruvate carboxykinase (ATP) 1. Some studies have described the importance of sugars in plant defense against biotic and abiotic stresses [65]. Carbohydrates can also provide materials for the synthesis of defense compounds [66]. A balance between glycolysis and gluconeogenesis was reported in *Q. ilex* seedlings in response to *P. cinnamomi* [29]. Beta-xylosidase/alpha-L-arabinofuranosidase 2 and beta-galactosidase are cell wall proteins that play crucial roles in wall modification and the defense response against fungi, and function in the context of water deficit [67]. Two thaumatin-like proteins (TLPs), TLP1 and TLP1a, were also common to both populations (Table 2). TLPs are pathogenesis-related-5 (PR-5) proteins involved in the stress response. Studies of in vitro antifungal activity in Picea likiangensis against the pathogenic fungus Lophodermium piceae provided strong evidence for the significant role of these proteins in defense [68]. Among the stress proteins, ornithine aminotransferase (OAT), a protein involved in proline synthesis, was also found to be accumulated in both populations. The accumulation of proline is a well-documented mechanism that improves water stress tolerance in plants, as evidenced by numerous studies [69,70]. 

Regarding specific proteins, more up-accumulated enzymes were identified in the Hu population than the Co population (92 vs. 76). The most abundant functional groups of proteins accumulated in Hu were transcription/translation; folding, sorting, and degradation; and carbohydrate metabolism. Among the transcription/translation group, we found two chloroplast ribosomal proteins already described in previous studies as being related to drought tolerance or the response to combined P cinnamomi–drought stress in holm oak: the 50S ribosomal proteins L17 and L22. Many of the 50S ribosomal proteins have been shown to be positively regulated in response to abiotic stress to compensate for damage to photosynthetic proteins in response to drought [34,71]. The 60S ribosomal proteins L6, L13-1, L13a-2, and L21-1 and ubiquitin-60S ribosomal protein L40 have also been shown to accumulate in response to the pathogen. Transcriptomic and proteomic analyses showed an increase in 60S ribosomal protein in *Q. ilex* seedlings in response to drought [72] and to combined *P. cinnamomi*–drought stress [34].

Among the folding, sorting, and degradation group, chaperones were the most represented. Heat shock proteins (HSPs) are ubiquitous proteins initially characterized in terms of their response to heat shock, but are now recognized to be induced by a wide range of stressors, including exposure to cold, UV light, wounds, tissue remodeling, and biotic stresses [73]. HSPs have also been related to biotic and abiotic stress response in *Q. ilex* [34,74]. Other chaperones identified were DnaJ A6, DnaK, ClpB, Cpn60, and T-complex protein 1 subunit beta. ClpB has already been found to increase in *Q. ilex* under drought conditions [74]. DnaJ proteins alone or in association with HSP70, such as chaperone protein DnaK, are involved in protective mechanisms to rapidly and effectively repair PSII damaged by environmental stress [75]. 

Transport proteins, such as TOC75 and the translocon on the outer membrane of the chloroplast 64 protein, were also found to be accumulated in the Hu population in response to the pathogen. The system comprising a translocon at the outer chloroplast envelope (TOC) and a translocon at the inner chloroplast envelope (TIC) mediates the import of the vast majority of nuclear-encoded plastid proteins from the cytoplasm [76]. The regulation of TOC–TIC activities is essential in the dynamic remodeling of the organelle proteome, which is required to coordinate plastid biogenesis with developmental and physiological events. Importing photosynthetic proteins and balancing the stoichiometry of nuclear- and chloroplast-encoded subunits of the multisubunit photosynthetic apparatus are critical for the maintenance and repair of the photosynthetic apparatus under stress [77]. In addition, the V-type proton ATPase, a proton pump that contributes to cytosolic pH homeostasis and energizes transport processes across endomembranes of the secretory pathway, is crucial in the response to abiotic and biotic stresses [78]. Two subunits of this protein (A and C) were identified to be involved in the response to *P. cinnamomi*, with the latter significantly more represented in the combined stress groups (Table 2). This protein was previously identified to be involved in *Q. ilex* in response to *P. cinnamomi* [34]. In addition, the carotenoid cleavage dioxygenase has an essential role in the response to osmotic stress [79].

Another less represented group of proteins found to be accumulated in the Hu population were stress response proteins snakin-2 and temperature-induced lipocalin-1. Snakins are host defense peptides that can inhibit the growth of various bacteria and fungi at low concentrations [80]. Lipocalins are photoprotective proteins induced in response to various abiotic stresses, including dehydration, high light, and low temperature. These proteins contribute to protecting the thylakoid membrane against oxidative damage, thereby preventing the accumulation of fatty acid hydroperoxides and lipid peroxidation [81,82]. Lipocalin along with betaine aldehyde dehydrogenase and chlorophyll a-b binding chloroplastic protein were increased in *Q. ilex* under drought conditions [35,57].

Specific proteins up-accumulated in the Co population belonged to the main functional groups of carbohydrate metabolism; folding, sorting, and degradation; transcription; and redox (Appendix A Table 2). Glyoxalase I and aconitate hydratase were accumulated in response to the combined stress. The glyoxalase (GLY) system, comprising GLYI and GLYII enzymes, has emerged as one of the primary detoxification pathways of methylglyoxal, a cytotoxic by-product of glycolysis, and is indispensable during abiotic and biotic stresses [83]. The role of glutathione (GSH) as an ROS scavenger regulating GLY enzymes and its physiological role in regulating various stress responses are well documented [84]. Aconitate hydratase, an enzyme of the glyoxylate bypass for the utilization of fatty acids as a carbon source, was identified in Castanea sativa in response to *P. cinnamomi* [85]. A delicate balance is required for proper functioning of this system, particularly during abiotic/biotic stress when the redox balance in cells is perturbed. In addition, several redox enzymes identified in this study, including glutathione reductase, thioredoxine reductase, NADPH-dependent thioredoxin reductase 3, monothiol glutaredoxin-S17, short-chain dehydrogenase/reductase SDR, and others, are also related to tolerance to drought [35,57] and resistance to *P. ciannamomi* [34] in *Q. ilex*.

The proteins of the transport porin/voltage-dependent anion-selective channel and GTP-binding nuclear protein Ran-3 were also accumulated in response to the pathogen, and the former was significantly accumulated under combined stress. Ion channels play an important role in turgor regulation in guard cells and become critical molecular targets of various signals, including osmotic stress [86,87]. Ras-related nuclear protein (RAN), a GTP-binding protein, functions in the nuclear–cytoplasmic transport of proteins and is involved in both the gibberellin and salicylic acid signaling pathways, promoting growth and disease resistance [88]. Mitochondrial Rho GTPase 1 belongs to the Rho family of proteins (ROPs), which are involved in plant immunity or susceptibility to disease [89]. The involvement of ROPs in plant–microbe interactions has been known for a long time. Activated immunogenic signals are transduced into characteristic immune reactions [89], as was demonstrated during the rice–*Magnaporthe oryzae* interaction [90,91]. Other proteins related to the defense response, such as protease stomatin/prohibitin-like protein, heat shock 70 kDa, carboxylesterase (CXE), and lipoxygenase (LOX), were also found to be accumulated in Co. Prohibitins and stomatins belong to a superfamily of proteins, together with a group of plant defense response genes named PID, referring to proliferation, ion, and death. Prohibitins are involved in proliferation and cell cycle control, and stomatins in ion channel regulation [92]. The role of CXE in response to pathogen attacks is well known [93], as was demonstrated in Arabidopsis transgenic plants, in which the overexpression of the *AtCXE8* gene led to enhanced resistance to *Botrytis cinerea* [94]. Finally, the involvement of LOX in the defense response against biotic and abiotic environmental stresses has also been described [95].

### 3.3. Metabolomic Analysis

A total of 13,236 (negative and positive) ion features were resolved for both populations and treatments; 4069 of them were considered to be consistently present in at least one condition (Appendix A), of which 1331 were annotated according to the metabolite databases (see Section 2.7). Two-dimensional PCA explained more than 50% of the total variability for the Co and Hu populations (Appendix A). PC1 discriminated between treatments, explaining 33.4% of the variability in Co and 36.4% in Hu. Qualitative and significant quantitative differences between treatments showed 508 and 611 up-accumulated metabolites for Co and Hu, respectively, and 633 and 542 down-accumulated metabolites. Focusing on significantly up-accumulated metabolites, the Venn diagram analysis showed 123 common to both populations (78 of them putatively annotated), 385 detected exclusively in Co (205 putatively annotated), and 488 in Hu (278 putatively annotated) (Figure 3d, Appendix A). 

A total of 97 metabolites (20 common to both populations, 31 specific to Co and 46 to Hu) were mapped to different pathways analyzed using the KEGG database (Appendix A). The amino acid biosynthesis pathway was the best represented, including non-aromatic and aromatic amino acid metabolism, followed by carbohydrate metabolism, and phenolics and secondary metabolism. Lipid and energy metabolism, aminoacyl-tRNA biosynthesis pathways, and the metabolism of cofactors and vitamins are also highlighted in the metabolic pathway enrichment analysis (Appendix A). Specific metabolites will be discussed in an integrated manner along with the enzymes of their metabolic pathways.

### 3.4. Integrated Proteome–Metabolome Analysis

An integrated analysis of proteins and metabolites up-accumulated in the Co and Hu populations in response to combined stress was performed. Pathway enrichment analysis revealed the most represented pathways in response to *P. cinnamomi* inoculation (Appendix A): carbohydrate metabolism (mainly represented by pyruvate metabolism, the citrate cycle, glycolysis/gluconeogenesis, and glyoxylate and dicarboxylate metabolism) with 41 enzymes and 13 metabolites; amino acid metabolism (alanine, aspartate, and glutamate metabolism; arginine and proline metabolism; phenylalanine, tyrosine, and tryptophan biosynthesis; glycine, serine, and threonine metabolism; and glutathione metabolism, among others) with 22 enzymes and 26 metabolites; energy metabolism, with 15 enzymes and 5 metabolites; and secondary metabolism, with 10 enzymes and 15 metabolites (Appendix A). A heat map showing the differential patterns of the functional groups identified in response to *P. cinnamomi*, based on the abundance of enzymes, was constructed (Figure 4). Two main clusters were generated: one included pathways more represented in the Hu population in response to *P. cinnamomi* (mainly amino acid metabolism and isoquinoline alkaloid biosynthesis), and the other included pathways more represented in the Co population (glycolysis/gluconeogenesis, terpenoid backbone biosynthesis, flavonoid biosynthesis, some amino acid metabolism). Several pathways were highly represented in both populations (pyruvate metabolism, the TCA cycle, phenylpropanoid biosynthesis, and aminoacyl-tRNA biosynthesis, among others). MOFA inferred three latent factors (F1, F2, and F3) (Appendix A). Overall, proteomics explained more variability than metabolomics (Appendix A). Significantly up-accumulated proteins and metabolites that presented high loadings (around the top 10% in absolute value) on the three factors are discussed and represented in Appendix A.

Enzymes and metabolites of the aminoacyl-tRNA biosynthesis pathway, such as leucine–tRNA ligase, glycine–tRNA ligase, methionyl-tRNA synthetase, L-phenylalanine, L-arginine, L-glutamate, and L-valine, were mainly accumulated in the Hu population. Methionyl-tRNA synthetase was previously related to drought tolerance in holm oak [57]. The association of tRNA transcription and modification with plant cell growth and the response to pathogens has been described. In addition, mutations in proteins directly involved in tRNA synthesis and modification most frequently lead to pleiotropic effects on plant growth and immunity [96]. 

Enzymes and related metabolites of carbohydrate and amino acid metabolism were accumulated in the Hu population in response to the combined stress, including alpha 1,4-glucan phosphorylase, phosphoglycerate kinase, 6-phosphofructokinase, pyrophosphate-dependent, aconitase, glutamate synthase (ferredoxin), D-mannose, citrate, L-malate, and L-glutamate. Some of these proteins have been related to tolerance to drought stress in *Q. ilex* [57], *Q. lobata*, and *Nitraria tangutorum* [97,98], and heat stress in *Populus tomentosa* [99]. In the Co population, enzymes and metabolites of the glycolysis/gluconeogenesis pathways, including fructose-1,6-bisphosphatase, pyruvate kinase, phosphoglycerate mutase, alcohol dehydrogenase, pyruvate, malate, citrate, and lactate, among others, were also represented. This can be explained as a consequence of defective photosynthetic activity in inoculated plants, as was reported in *Q. ilex* in response to *P. cinnamomi* by Sghaier-Hammami et al. [29].

Other metabolic pathways also increased in response to the combined stress, such as the secondary metabolism pathways. The enzymes of the phenylpropanoid biosynthesis pathway, cinnamyl alcohol dehydrogenase (EC1.2.1.44), cinnamyl alcohol dehydrogenase 1 (EC1.1.1.195), and the metabolites coumarin and 4-coumaryl alcohol, were up-accumulated in the Co population. In addition, other enzymes and metabolites of flavonoid biosynthesis, including chalcone synthase, catechin, piceol, and silybin, were also increased in response to the pathogen (Appendix A). The increase in the phenylpropanoid pathway can be interpreted as a mechanism of defense by strengthening the cell wall by incorporating lignin through the activity of cinnamyl alcohol dehydrogenases [100], as was proposed in *Q. pubescens* leaves in response to drought [101] and in *Q. variabilis* in response to *P. cinnamomi* [32]. Flavonoid biosynthesis has also been linked to tolerance to drought in *Q. ilex* [57,102] and the response to *P. cinnamomi* in *C. sativa* and *Quercus* spp. [32,85]. 

Overall, most of the proteins identified in response to the pathogen under watering conditions were also found in the presence of the pathogen under non-watering conditions. Therefore, a differential response was not observed between the two conditions of inoculation, as was previously observed in other holm oak populations [34]. There appears to be a protective effect of the photosynthetic apparatus and the osmotic regulation mechanism in both populations as a consequence of defective photosynthetic activity in inoculated plants. The activation of the glyoxalase system, together with a battery of ROS enzymes, may provide a more efficient detoxification mechanism for Co than Hu. This is consistent with the decreased leaf chlorophyll fluorescence observed on inoculated plants, which was more apparent in the Hu population. 

## 4. Conclusions

Based on physiological and molecular analysis, the defense response to *P. cinnamomi* is dependent on the population, with the Co population found to be more resistant than the Hu population. No significant differences were observed between watered and non-watered inoculated plants, although the latter condition caused more severe symptoms. Proteomic and metabolomic analysis revealed differential involvement of metabolic pathways in both populations in response to *P. cinnamomi*. The accumulation of enzymes and metabolites of the glyoxalase system of detoxification, ROS proteins, and the phenylpropanoid and flavonoid biosynthesis pathways may provide higher resistance for the Co population through a more efficient system of protecting the photosynthetic apparatus and strengthening of the cell wall through the incorporation of lignin as a mechanism of defense. Some proteins related to pathogen defense were identified, especially in the Co population. However, overall, the observed response resembled that seen in response to drought. The increased levels of some specific enzymes and metabolites in the Co population may be key to providing greater resistance to the pathogen. This information highlights the variability in the response of holm oak to *P. cinnamomi*. Some candidate genes, among them beta-galactosidase, ornithine aminotransferase, glyoxalase I, glutathione reductase, thioredoxin reductase, and cinnamyl alcohol dehydrogenase, could be used for the selection of elite genotypes in breeding programs.

## Figures and Tables

**Figure 1 biomolecules-14-00160-f001:**
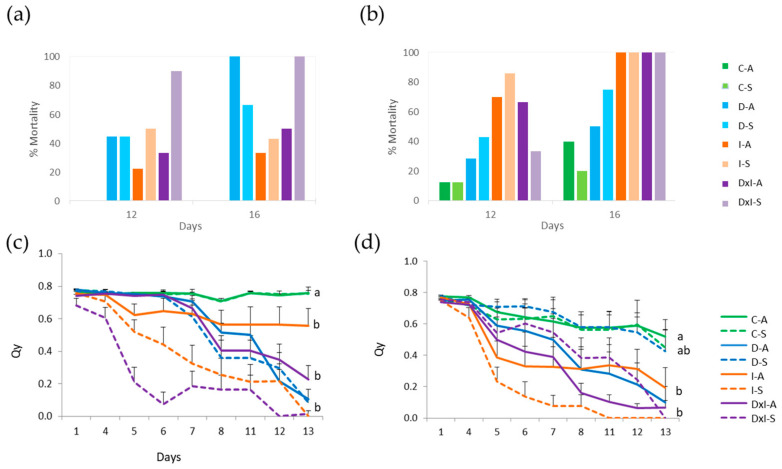
(**a**,**b**) Percentage of seedling mortality and (**c**,**d**) measurements of quantum yield of photosystem II (Qy) in dark-adapted leaves from Cordoba (**a**,**c**) and Huelva (**b**,**d**) populations during drought (D), *P. cinnamomi* inoculation (I), and combined (D × I) stress with respect to control (C) seedlings from asymptomatic (A) and symptomatic (S) *Q. ilex* mother trees. Values represent mean ± SE of three biological replicates. Different letters, calculated from the area under the Qy curve, denote significant differences among treatments (*p* < 0.05).

**Figure 2 biomolecules-14-00160-f002:**
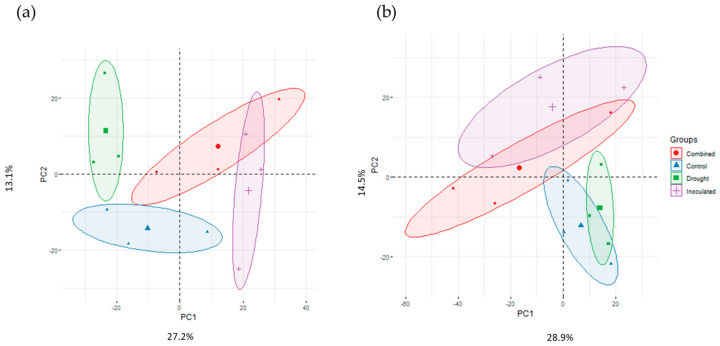
Principal component analysis of proteins consistently identified in seedlings from asymptomatic (**a**) Cordoba and (**b**) Huelva mother trees under different treatments: control, drought, inoculation, and combined stress.

**Figure 3 biomolecules-14-00160-f003:**
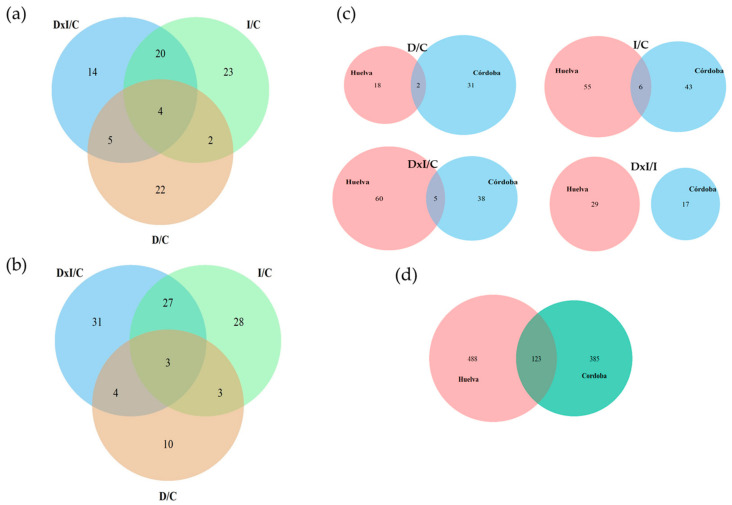
Variable proteins significantly up-accumulated in seedlings from asymptomatic (**a**) Cordoba and (**b**) Huelva mother trees under different treatments: control (C), drought (D), inoculation (I), and combined stress (D × I). (**c**) Comparative analysis of both populations in response to different stresses. (**d**) Variable metabolites significantly up-accumulated under combined stress.

**Figure 4 biomolecules-14-00160-f004:**
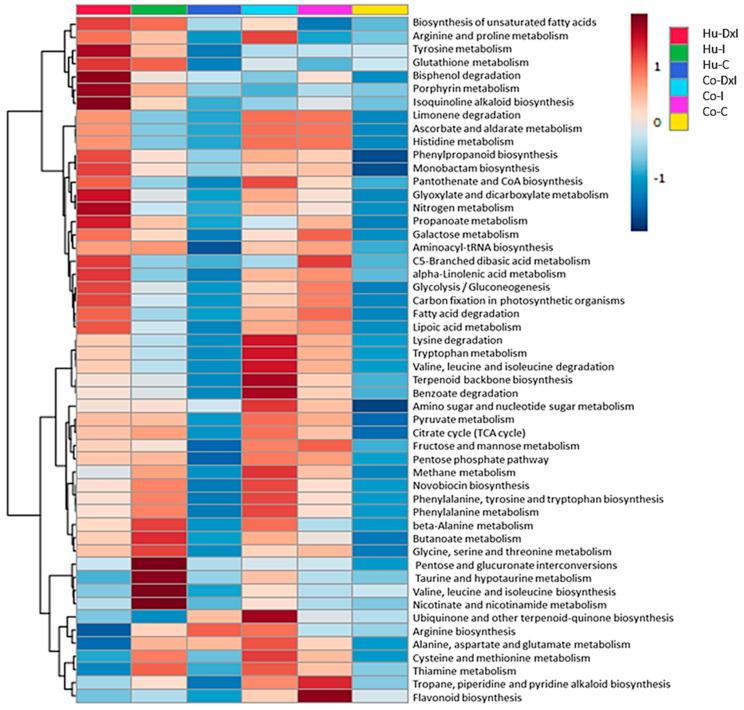
Heat map showing differential patterns of functional groups of protein in response to *P. cinnamomi* with (I) and without (D × I) watering, and in control (C) group. Co (Cordoba) and Hu (Huelva) populations.

**Table 1 biomolecules-14-00160-t001:** Summary of features of shotgun proteomic/metabolomic analysis.

Feature	Cordoba	Huelva	Common to Both
Raw data	2735/10,983	3019/10,051	2540/7798
≥2 peptides	2261	2451	2261
Score ≥ 2	2260	2448	2203
Consistent proteins	1590	1444	1334
Statistical significance (p ≤ 0.05)	600/1298	525/1309	350/503
Up-accumulated:			
D/C	33	20	2
I/C	49	61	6
DxI/C	43/385	65/488	5/123
DxI/I	17	29	0

**Table 2 biomolecules-14-00160-t002:** Significantly up-accumulated proteins classified by functional group in response to drought (D), *P. cinnamomi* inoculation (I), and combined stress (D × I) compared to control seedlings from asymptomatic Cordoba (Co) and Huelva (Hu) mother trees. Asterisks indicate up-accumulated proteins in D × I compared to I.

Molecular Function	Protein ID	Protein Description	Up-Accumulated	Population
Carbohydrate metabolism	qilexprot_70617	Fructose-1,6-bisphosphatase, chloroplastic	D × I/D × I, I	Co/Hu
	qilexprot_74362	Beta-xylosidase/alpha-L-arabinofuranosidase 2	D√I/I	Co/Hu
	qilexprot_27855	D-tagatose-1,6-bisphosphate aldolase subunit gatY	D × I, I/D × I, I	Co/Hu
	qilexprot_67493	Phosphoglycerate mutase (2,3-diphosphoglycerate-independent)	D × I, I/I	Co/Hu
	qilexprot_46285	Phosphoenolpyruvate carboxykinase (ATP) 1	D × I, I, D/D × I, I	Co/Hu
	qilexprot_11680	Pyruvate dehydrogenase E1 component subunit alpha, mitochondrial	I/D × I	Co/Hu
	qilexprot_45099	Beta-galactosidase	I/D × I, I	Co/Hu
	qilexprot_65736	Phosphoglycerate kinase	D	Co
	qilexprot_54727	Malate dehydrogenase	D	Co
	qilexprot_57046	Malate dehydrogenase	D × I	Co
	qilexprot_32059	Glucose-6-phosphate 1-dehydrogenase, cytoplasmic isoform 2	D	Co
	qilexprot_19946	Pyruvate dehydrogenase E1 component subunit beta, mitochondrial	D	Co
	qilexprot_10831	FGGY carbohydrate kinase domain-containing protein	D	Co
	qilexprot_58277	Aconitate hydratase	D × I	Co
	qilexprot_73875	Malic enzyme	D × I, D	Co
	qilexprot_20787	Uridine 5’-diphosphate-sulfoquinovose synthase	D × I, I	Co
	qilexprot_43884	Glyoxalase I	D × I, I	Co
	qilexprot_52600	Pyruvate kinase	D × I, I	Co
	qilexprot_12453	Fructose-1,6-bisphosphatase	D × I, I	Co
	qilexprot_32875	Alcohol dehydrogenase 1	D × I, I	Co
	qilexprot_48598	Pyruvate decarboxylase 1	D × I, I, D	Co
	qilexprot_38653	Succinate dehydrogenase (ubiquinone) flavoprotein subunit, mitochondrial	I	Co
	qilexprot_18792	Triosephosphate isomerase, cytosolic	I	Co
	qilexprot_38624	ATP-dependent 6-phosphofructokinase	I	Co
	qilexprot_19665	Ribokinase	I	Co
	qilexprot_61660	Sorbitol dehydrogenase	I	Co
	qilexprot_49147	Succinyl–CoA ligase (ADP-forming) subunit alpha	I, D	Co
	qilexprot_48238	Granule-bound starch synthase 1, chloroplastic/amyloplastic	D	Hu
	qilexprot_27919	Alpha 1,4-glucan phosphorylase	D × I	Hu
	qilexprot_55364	Glyceraldehyde-3-phosphate dehydrogenase	D × I	Hu
	qilexprot_75569	Glyceraldehyde-3-phosphate dehydrogenase	D × I	Hu
	qilexprot_4907	Probable galactinol-sucrose galactosyltransferase 2	D × I, I	Hu
	qilexprot_49174	6-Phosphogluconate dehydrogenase, decarboxylating	D × I, I, D	Hu
	qilexprot_16282	Aconitase	I	Hu
	qilexprot_178	Acetyltransferase component of pyruvate dehydrogenase complex	I	Hu
	qilexprot_50083	Phosphoglycerate kinase	D × I	Hu
	qilexprot_46354	Succinyl–CoA ligase (ADP-forming) subunit alpha	D × I	Hu
	qilexprot_7036	6-phosphofructokinase, pyrophosphate-dependent	I	Hu
Amino acid metabolism	qilexprot_49163	Aspartate aminotransferase 3, chloroplastic	D/D	Co/Hu
	qilexprot_19886	Phosphoserine aminotransferase 1, chloroplastic	D/D × I, I	Co/Hu
	qilexprot_7937	Glutamine synthetase	D × I	Co
	qilexprot_968	Ferredoxin-dependent glutamate synthase	D × I, I	Co
	qilexprot_73896	Carbamoyl-phosphate synthase large chain, chloroplastic	D × I, I	Co
	qilexprot_49389	Arginase 1, mitochondrial	D	Hu
	qilexprot_74641	D-3-phosphoglycerate dehydrogenase 1, chloroplastic	D × I	Hu
	qilexprot_41437	Glutamate synthase (ferredoxin)	D × I, I	Hu
	qilexprot_45599	Glutamate synthase (ferredoxin)	D × I, I	Hu
	qilexprot_26928	Glutamate synthase (ferredoxin)	D × I, I	Hu
	qilexprot_7419	Serine hydroxymethyltransferase	I	Hu
	qilexprot_55598	Alpha isopropylmalate synthase	I	Hu
Lipid metabolism	qilexprot_55785	Epoxide hydrolase 3	D	Co
	qilexprot_56587	NADH-cytochrome b5 reductase-like protein	D	Co
	qilexprot_36359	Acyl-CoA oxidase 1	D	Co
	qilexprot_19906	Acyl-CoA dehydrogenase	D	Co
	qilexprot_58843	Acyl-coenzyme A oxidase	D	Co
	qilexprot_29429	Acetyl-CoA acetyltransferase	D × I	Co
	qilexprot_66905	CXE carboxylesterase	I	Co
	qilexprot_40937	CXE1 Carboxylesterase 1	I	Co
	qilexprot_72077	Lipoxygenase *	D × I	Co *
	qilexprot_60909	12-Oxophytodienoate reductase 1	D	Hu
	qilexprot_77659	Phospholipase D alpha 1	D × I	Hu
	qilexprot_61680	3-Hydroxybutyryl-CoA dehydratase	I	Hu
	qilexprot_10167	Acetate/butyrate–CoA ligase AAE7, peroxisomal	I, D	Hu
Nucleotide metabolism	qilexprot_35697	Phosphoribosylaminoimidazolecarboxamide formyltransferase	D × I, I	Co
	qilexprot_38489	Dihydropyrimidine dehydrogenase (NADP(+)), chloroplastic	D × I	Hu
	qilexprot_29525	Adenylosuccinate synthetase, chloroplastic	D × I, I	Hu
Cellular processes	qilexprot_59288	MPBQ/MSBQ methyltransferase	D × I	Co
	qilexprot_72324	Putative membrane protease, stomatin/prohibitin-like protein	D × I, D	Co
	qilexprot_26311	Aluminum-induced protein with YGL and LRDR motifs	I	Co
	qilexprot_32288	Dynamin-related protein 3A	D × I	Hu
	qilexprot_19558	Dynamin-related protein 3A	D × I, I	Hu
	qilexprot_31321	Carotenoid cleavage dioxygenase 1	D × I	Hu
	qilexprot_9244	Turgor-responsive-like protein	D × I, I	Hu
	qilexprot_16156	Tublin binding cofactor C	I	Hu
Development	qilexprot_67941	Apoptosis inhibitor 5-like protein API5	I	Hu
	qilexprot_56133	ATPase family AAA domain-containing protein 3	I	Hu
	qilexprot_68348	Sieve element occlusion protein 1	I	Hu
Photosynthesis	qilexprot_6646	Chlorophyll a-b binding protein, chloroplastic	D	Co
	qilexprot_16571	Chlorophyll a-b binding protein, chloroplastic	D	Hu
	qilexprot_77212	Cytochrome b559 subunit alpha	D	Hu
Energy metabolism	qilexprot_2302	Ferredoxin-NADP reductase	I	Hu
Folding, sorting and degradation	qilexprot_41199	Small heat shock protein, chloroplastic	I/D × I, I	Co/Hu
	qilexprot_73457	Non-heme dioxygenase N-terminal domain-containing protein	D	Co
	qilexprot_24541	Non-heme dioxygenase N-terminal domain-containing protein	D × I, D	Co
	qilexprot_10667	Glycine cleavage system H protein, mitochondrial	D × I	Co
	qilexprot_22106	Xaa-Pro aminopeptidase 2	D × I	Co
	qilexprot_76191	Heat shock 70 kDa protein	D × I, I	Co
	qilexprot_68090	Putative heat shock protein 70	I	Co
	qilexprot_42055	Proteasome subunit beta type-1	I	Co
	qilexprot_4991	Heat shock protein 40	D	Hu
	qilexprot_11703	Chaperone protein DnaJ A6, chloroplastic	D × I	Hu
	qilexprot_17542	22.0 kDa class IV heat shock protein	D × I	Hu
	qilexprot_67032	Proteasome subunit beta type 4	D × I	Hu
	qilexprot_36038	Small heat shock protein	D × I	Hu
	qilexprot_55307	Small heat shock protein, chloroplastic	D × I, I	Hu
	qilexprot_29045	Protein Cpn60 *	D × I	Hu/Co *
	qilexprot_38494	Presequence protease 1, chloroplastic/mitochondrial	D × I, I	Hu
	qilexprot_31991	Oligopeptidase A (partial)	D × I, I	Hu
	qilexprot_50467	ClpB	D × I, I	Hu
	qilexprot_46770	TCPB	D × I, I	Hu
	qilexprot_46485	Chaperone protein DnaK	D × I, I	Hu
	qilexprot_55829	T-complex protein 1 subunit beta	D × I, I, D	Hu
	qilexprot_68821	Prolyl tripeptidyl peptidase	I	Hu
Redox	qilexprot_41994	Oxidoreductase	D	Co
	qilexprot_47035	NADH-ubiquinone reductase 75 kDa subunit	D × I	Co
	qilexprot_8286	Monothiol glutaredoxin-S17	D × I	Co
	qilexprot_22755	Thioredoxin reductase	D × I, I	Co
	qilexprot_13916	Glutathione reductase, chloroplastic	D × I, I	Co
	qilexprot_4433	NADPH-dependent thioredoxin reductase 3	D × I, I	Co
	qilexprot_27576	Short-chain dehydrogenase/reductase SDR	I	Co
	qilexprot_67856	Sulfite reductase 1 (ferredoxin), chloroplastic	I	Co
	qilexprot_53092	Thioredoxin reductase 1	I	Co
	qilexprot_38487	Glutathione reductase	I	Co
	qilexprot_21080	Putative uncharacterized protein PDI	D × I	Hu
	qilexprot_21353	Mitochondrial peroxiredoxin IIF	D × I	Hu
	qilexprot_67058	Aldo/keto reductase	D × I	Hu
	qilexprot_6108	Peroxiredoxin-2B	D × I	Hu
	qilexprot_4209	Polyphenol oxidase	D × I, D	Hu
	qilexprot_45306	NADPH HC toxin reductase	D × I, D	Hu
Secondary metabolism	qilexprot_7930	Cinnamoyl alcohol dehydrogenase	D	Co
	qilexprot_62609	Clavaminate synthase-like protein	D	Co
	qilexprot_67426	Cinnamyl alcohol dehydrogenase 1	D × I, I	Co
	qilexprot_29542	Chalcone synthase	I	Co
	qilexprot_16776	Protein STRICTOSIDINE SYNTHASE-LIKE 3	D	Hu
	qilexprot_29588	Betaine aldehyde dehydrogenase	I	Hu
	qilexprot_6949	Glutamate decarboxylase	I	Hu
Stress	qilexprot_16524	Ornithine aminotransferase	D/D × I, D	Co/Hu
	qilexprot_66333	Thaumatin-like protein 1a	D × I, D/D × I, I	Co/Hu
	qilexprot_52513	Thaumatin-like protein 1	D × I, D/D × I, I	Co/Hu
	qilexprot_62485	Snakin-2	D × I	Hu
	qilexprot_4320	Temperature-induced lipocalin-1	D × I	Hu
	qilexprot_75875	Cysteine desulfurase 1, chloroplastic	I	Hu
	qilexprot_77543	Succinate-semialdehyde dehydrogenase	I	Hu
Transcription/Translation	qilexprot_49279	Plastid lipid-associated protein 3, chloroplastic	I/D × I, I	Co/Hu
	qilexprot_32535	Plastid transcriptionally active 16	D × I, I	Co
	qilexprot_33379	Plastid transcriptionally active 16	D × I, I	Co
	qilexprot_40661	Histone H2A	D × I, I	Co
	qilexprot_59253	Histone H2A	I, D	Hu
	qilexprot_31771	EF-Tu	D × I, I	Co
	qilexprot_48469	DNA-directed RNA polymerase subunit alpha	D × I, I, D	Co
	qilexprot_70773	Alanine–tRNA ligase	I	Co
	qilexprot_67436	50S ribosomal protein L1, chloroplastic	I	Co
	qilexprot_61172	Ubiquitin-conjugating enzyme E2 variant 1D	D	Hu
	qilexprot_35357	Developmentally regulated G-protein 3	D × I	Hu
	qilexprot_47135	50S ribosomal protein L17, chloroplastic	D × I	Hu
	qilexprot_28988	Elongation factor 2	D × I	Hu
	qilexprot_48468	30S ribosomal protein S8, chloroplastic	D × I	Hu
	qilexprot_22027	Histone H4	D × I	Hu
	qilexprot_71551	Ubiquitin-60S ribosomal protein L40	D × I	Hu
	qilexprot_16675	50S ribosomal protein L22, chloroplastic	D × I, D	Hu
	qilexprot_74256	Leucine–tRNA ligase, cytoplasmic *	D × I, I	Hu *
	qilexprot_42234	Glycine–tRNA ligase, chloroplastic/mitochondrial 2	D × I, I	Hu
	qilexprot_49297	60S ribosomal protein L13-1	I	Hu
	qilexprot_19368	Polyadenylate-binding protein RBP45	I	Hu
	qilexprot_50137	60S ribosomal protein L13a-2	I	Hu
	qilexprot_38968	60S ribosomal protein L21-1	I	Hu
	qilexprot_56118	60S ribosomal protein L6	I	Hu
	qilexprot_46161	Methionyl-tRNA synthetase	D × I, I	Hu
Signaling	qilexprot_22041	Mitochondrial Rho GTPase 1	D × I	Co
	qilexprot_41913	NADH:ubiquinone oxidoreductase complex I intermediate-associated protein 30	D × I, I	Co
Transport	qilexprot_13099	Porin/voltage-dependent anion-selective channel protein *	D × I, I, D	Co */Hu *
	qilexprot_22308	GTP-binding nuclear protein Ran-3	I	Co
	qilexprot_20347	Hexapeptide transferase family protein	I	Co
	qilexprot_25532	Ferritin	D × I	Hu
	qilexprot_9762	Nuclear pore complex protein NUP98A	D × I, I	Hu
	qilexprot_48381	Clathrin heavy chain 2	D × I, I, D	Hu
	qilexprot_19028	Polyadenylate-binding protein 8 *	I	Hu/Co *
	qilexprot_23804	Protein TOC75, chloroplastic	I	Hu
	qilexprot_16395/ qilexprot_4497	V-type proton ATPase catalytic subunit A/V-type proton ATPase subunit C *	I	Hu *
	qilexprot_64280	Translocon at outer membrane of chloroplasts 64	I	Hu
	qilexprot_74226	PABP	I	Hu
Miscellaneous	qilexprot_29442	Aldehyde dehydrogenase *	D	Co *
	qilexprot_42491	Citrate synthase	D	Co
	qilexprot_71234	UDP-sulfoquinovose synthase	D × I	Co
	qilexprot_14071	At2g30880/F7F1.9 *	I, D	Hu *
Not assigned	qilexprot_79169	Putative uncharacterized protein Sb03g011745	D	Co
	qilexprot_20701	Putative reversibly glycosylatable polypeptide	D	Co
	qilexprot_48145	Putative uncharacterized protein At5g13030	D × I	Co
	qilexprot_36527	Fruit protein pKIWI502	I, D	Co
	qilexprot_45328	NAD(P)-binding Rossmann-fold protein	D	Hu
	qilexprot_34141	NAD(P)-binding Rossmann-fold protein	D × I	Hu
	qilexprot_72624	Spermatogenesis-associated protein 20	D × I, I	Hu
	qilexprot_63727	Pdx1 *	D × I	Hu *

## Data Availability

Data are available in the repositories described in the manuscript and in the Appendix A.

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
