# Peer review of "Proteomic and Metabolomic Analysis of the Quercus ilex–Phytophthora cinnamomi Pathosystem Reveals a Population-Specific Response, Independent of Co-Occurrence of Drought"

_biomolecules, 2024, doi:10.3390/biom14020160_

Round 1

Reviewer 1 Report

Comments and Suggestions for Authors

Dear Authors,

I have reviewed your manuscript "Proteomic and metabolomic study of the Quercus ilex response to Phytophthora cinnamomi, the main causal agent of the decline syndrome, and the identification of putative markers of resistance", submitted for publication in Biomolecules.

I find that your manuscript is well-structured and mostly well-written. The research design is comprehensive and appropriate, and the results are appropriately discussed. The paper presents several problems such as imperfect English language, limited clarity of certain points (especially due to the complex experimental design, which, at times, leaves the reader unclear as to which experimental treatment or genotype you are referring to), and insufficiently profiled summarization and wrap-up of the main message of the paper. For particular suggestions for improvement, please follow my comments as given below:

·         Manuscript title - I find that your manuscript title is too long and too vague. In the recent years, vague paper titles such as "proteomic and metabolomic study" or "identification of markers" are being phased out, giving room to more specific wording (for example: "Integrated proteomic-metabolomic responses of holm oak (Latin name) to Phytophthora cinnamomi, the causative agent of the oak decline syndrome, are population-specific but independent on co-occurrence of drought in Andalusian agroforestry ecosystems" or something similar - I am not specifically suggesting this title but rather giving an example of what I think an efficient title should sound like). On the other hand, the title of your paper is either way too long to keep efficiently the reader's attention, and you should consider cutting it down to 20 or at least around 25 words, if possible. This can be achieved through deleting the words "Phytophthora cinnamomi, the causative agent of" and leaving just "the oak decline syndrome" or in some other similar way.

·         English language - In the current version of your paper, English language suffers from moderate imperfections, remnants of sentence structures or even text written in your mother tongue. I am listing some examples below, but would recommend a thorough language editing by a first-language speaker scientist.

o    together WITH - at various places throughout your manuscript, you have omitted the word "with" from "together with" (lines 15, 35, 57; please check throughout the entire manuscript)

o    Sentence structure - at several points throughout the manuscript, sentences are structured in a way which is not appropriate for English language ("being the presence of the pathogen confirmed" - line 286; "being 350 common to both" - line 380-381; "such as was demonstrated... and so on to the end of the sentence" in line 497-499; "being more resistant the offspring... and so on to the end of the sentence" in line 603-604. Please also thoroughly check English grammar throughout the rest of the manuscript as well.)

o    Other:

§  line 15: please add the word "pathogens" ("one of the most aggressive pathogens")

§  line 67-68: "intra e inter population" - please revise

§  line 107: healthy acorns were surface cleanED

§  line 108: the floating acorns were discarded (not "those")

§  line 115: Germany

§  line 116: please replace "y" with "and"

§  line 137: < 2 mm thick (not "think")

§  line 210: All analyses (plural)

§  line 219: Please always use the capital letter L for "liter" (please also check elsewhere throughout the manuscript text)

§  line 287-288: "from trees both showing and not showing damage symptoms"

§  line 299: impredictable

§  line 310: I think it should be "PARPBH" instead of "PARBH", please double-check and revise if necessary

§  line 319: proving (or maybe better "suggesting")

§  line 320: of the latter (two letters T)

§  line 329: experiment (letter N is missing)

§  line 344-345: there were no differences (not "there were NOT differences")

§  line 452: was previously identiFIED

§  line 459-460: These proteins contribute (please delete the S at the end of "contribute")

§  line 498: Since AtCXE8 is a gene, its name should be written in italic letters

·         Experimental design: Your experimental design is quite complex, which is at times confusing to the reader, and the reader needs to go back and forth through your manuscript to make sure to understand all the different samples. Briefly, you used 16 different types of samples: 2 locations (Cordoba and Huelva) x 2 types of mother trees (infected with Phytophthora, i.e., symptomatic, and non-infected, i.e., asymptomatic) x 4 treatments (control, drought, infection with Phytophthora, DxI). While this variety of different types of samples makes your results more rich, and encompasses a greater collection of sources of variability of the seedling resistance (or susceptibility) traits, the complexity of this collection of samples may act confusing on the reader unless you minimize the room for confusion. I will now give you some ideas how to do that:

o    Abstract - lines 17-18: Please specify, already within the Abstract, that acorns were collected from both declined and non-declined trees at both locations (Cordoba and Huelva). Reading the Abstract, I immediately assumed that Cordoba was an area generally affected by oak decline and that Huelva was a healthy location - although you indeed did not use the word "respectively" - it made sense that you chose two different locations because one was affected by the oak decline disease and the other was not. This potential source of confusion should be prevented.

o    Introduction - a similar intervention is needed in the line 97 of the Introduction.

o    Materials & Methods - Although the entire experimental setup is properly explained within the section 2.2 (in lines 125-127 you clearly state that you had collected acorns from both healthy and infected trees at both Cordoba and Huelva locations), this point should also be more clearly indicated in the section 2.1, because this is where you describe the trees from which the acorns had been collected. Thus, at the beginning of the section 2.1, you should point out that you collected acorns from both healthy (asymptomatic) and infected (symptomatic) trees at both Cordoba and Huelva locations. Further confusion is added by mentioning that you had surface-cleaned the healthy acorns (line 107). Of course, I get that you only collected the healthy-looking acorns (even from the infected trees) for the purpose of the experiment. However, this needs to be emphasized more clearly to avoid confusing the reader.

o    line 161 - the proteins were isolated only from the leaves of seedlings derived from healthy mother plants, which means that for this analysis, you used 8 instead of 16 sets of samples (as also verified in Figures 2-5 as opposed to Figure 1). This is perfectly justified, but please, emphasize here more explicitly (in line 161) that for further proteomic and metabolomic analyses you used only half of your samples (i.e., only those germinated from acorns derived from asymptomatic mother trees) and provide a brief explanation as to why you decided to do so. This should be intended only to help the reader better grasp what are the results that he's looking at, and which analyses have been performed with which sets of samples, and why. The same brief statement (just that the results refer to the seedlings derived from acorns collected from asymptomatic trees) should be added to the Figure captions to all the Figures from Figure 2 to Figure 5 - again to reduce the room for confusion on the part of the reader when looking at the Figures.

o    line 321-324: Please be more specific here, about what you mean by "infected" (I presume it means "experimentally infected", as opposed to "derived from infected, i.e., symptomatic mother tree"). Because you mention both the experimentally infected seedlings, and their provenance from naturally infected mother plants, this passage of text can easily confuse the reader, unless you take measures to avoid that, by specifying in minute detail what you mean by each part of the sentence. Please also check other similar passages of text keeping in mind that although you know your experimental design very well, you may confuse the reader by mentioning experimentally infected seedlings and their naturally infected mother plants in closely connected parts of text.

o    line 158-159: Another clarification regarding your experiment, is needed here, when you mention that proteomics were performed "when leaf chlorophyll fluorescence decreased 35% (day 12 of the experiment)". This sentence is completely unclear for multiple reasons:

§  What was the main criterium for sampling for proteomics - sampling at day 12 (when leaf chlorophyll fluorescence happened to decrease to 35%), or sampling when leaf chlorophyll fluorescence decreased to 35% (which happened to arrive at day 12)?

§  In which set of plants did the leaf chlorophyll fluorescence drop to 35% at day 12? (I am sure it did not drop to 35% on the same day for both infected and non-infected plants, both those subjected and those not subjected to drought)

§  What does "decreased 35%" mean? "Decreased by 35% of the initial value (i.e., decreased to 65% the initial value), or "decreased to 35% of the initial value"?

§  A similar explanation is needed at line 305-306, when you mention again the same information.

o    line 327: Why do you call day 12 "the earliest time evaluated"? I have no record that you performed any measurements at some later point in time - unless I am terribly wrong.

·         Presentation of results - I have several important remarks regarding the presentation of your results, please see below:

o    Figure 1 - standard errors and statistical tests should be shown for the histograms in Figure 1A and B.

o    Figure 3 - In Figure 3, the "up-accumulated" and "down-accumulated" proteins for all the three experimental treatments (D, I, DxI) seem to be pooled together. This approach to the presentation of differentially expressed proteins does not make sense to me. If there is considerable overlap between the proteins that are up-accumulated or down-accumulated compared to control in response to all the three treatments, then that population of differentially expressed proteins might be interesting to show; but the way I understand the current version of Figure 3 does not really make sense. I suggest that Figure 3 should be removed or thoroughly revised.

o    line 395-396: What do the terms "specific" and "common" in these sentence refer to? "Specific" for a specific stress treatment and common to all the three stress treatments, or...? Please elaborate within the manuscript text.

o    line 408-409: Where do the TLPs come from now? I did not see them mentioned among the 14 common protein groups. I do not grasp the context of this sentence.

o    line 412-413: Is this a result from your paper, or from some previous work? Again, I do not see the context of this sentence.

o    Logically, Figure 4 should be presented before Table 2

o    Table 1 and Figure 4: While pairwise comparison of D (drought) or I (infection) to control is perfectly justified, I do not see why the combined treatment (DxI) should be compared to the control treatment. Comparison of DxI to I instead of DxI to control would make much more sense, because the DxI treatment was performed to determine in what way drought additionally affects (worsens) the (basal) treatment of infection. Thus, in scientific sense, you performed the DxI treatment to be able to compare it to I, not to control. I suggest that the comparisons DxI to C at the bottom of Table 1, as well as in the Venn diagrams in Figure 4, should be replaced with comparisons DxI to I.

·         Other remarks:

o    line 105: At what time of the year (and also, at which year) were the acorns collected?

o    line 108-109: the acorns were "extensively washed again" - washed with water? Did you use sterile tap water, distilled water, deionized water? Please be specific.

o    line 132: in sterile CA liquid, I presume? Please specify

o    Supplementary Figure 2B - a scale bar is mentioned in the caption to the supplementary figure, but is missing from the figure itself.

o    line 315: please add the word "considerably" ("showed considerably less mortality")

o    line 370-371: Please revise: "Two-dimensional PCA explained 40% and 43%..." - as PCA always explains the entire (100%) variability of the measured data, but since we only have a two-dimensional paper to visualize PCA, we usually display only a two-dimensional PCA in a diagram which explains only the two most contributing components of variability (PC1 + PC2). A similar addition to the text is needed again in line 510.

o    line 372: Please revise: "PC1 (27.2%) correlated to some extent with control, drought and inoculation treatments..."

o    line 508-518: You keep talking about "ions" in this paragraph, whereas the figures and tables that you cite in the text do not refer to compounds that are specifically ionized, but rather to various collections of metabolites. Please double-check the need to call these metabolites "ions" and either revise, or explain why you call them "ions" here all of a sudden.

·         Conclusion: Although you have quite successfully summarized the results of your research in the Conclusions section, I feel like a "future perspective" dimension is missing. A brief commentary about how your results might lay a foundation for future endeavors in fighting against Ph. cinnamomi is missing. I believe that such commentary would provide a proper wrap-up and conclusion to your article.

Despite my numerous comments and suggestions for correction, I want to emphasize that I appreciate your work very much and that you are presenting very important results in this work. You are working on an important topic that is supposed to importantly contribute to the protection of a vulnerable natural environment in the Mediterranean basin and for that matter I have great respect and admiration for your work. I hope that my comments will help you present your results in a more successful way and make greater scientific impact, with which I sincerely wish you the best of luck. I am looking forward to reading your published paper in Biomolecules.

Kind regards,

Reviewer

Comments on the Quality of English Language

·         English language - In the current version of your paper, English language suffers from moderate imperfections, remnants of sentence structures or even text written in your mother tongue. I am listing some examples below, but would recommend a thorough language editing by a first-language speaker scientist.

o    together WITH - at various places throughout your manuscript, you have omitted the word "with" from "together with" (lines 15, 35, 57; please check throughout the entire manuscript)

o    Sentence structure - at several points throughout the manuscript, sentences are structured in a way which is not appropriate for English language ("being the presence of the pathogen confirmed" - line 286; "being 350 common to both" - line 380-381; "such as was demonstrated... and so on to the end of the sentence" in line 497-499; "being more resistant the offspring... and so on to the end of the sentence" in line 603-604. Please also thoroughly check English grammar throughout the rest of the manuscript as well.)

Author Response

A letter to the reviewer has been attached.

Reviewer 2 Report

Comments and Suggestions for Authors
The manuscript focuses on the significance of holm oak (Quercus ilex) as a key component in Mediterranean forests and the Spanish "dehesa," highlighting its role in ecological and socio-economic sustainability. The research investigates the impact of the aggressive pathogen Phytophthora cinnamomi, coupled with drought, as major contributors to holm oak decline. The study examines the responses to P. cinnamomi inoculation in offspring from declining and non-declining areas of two Andalusian populations (Cordoba and Huelva), considering factors such as damage symptoms, mortality, and chlorophyll fluorescence under varying humidity conditions.
Results indicate population-dependent effects, with Huelva showing more pronounced responses than Cordoba. The integrative proteomic and metabolomic analysis identifies distinct metabolic pathways involved in the response to the pathogen, including amino acid metabolism in Huelva and terpenoids and flavonoids biosynthesis in Cordoba. Interestingly, no differential response is observed between seedlings inoculated under humid and drought conditions.
The study unveils a protective mechanism within the photosynthetic apparatus in response to impaired photosynthetic activity, notably more efficient in the Cordoba population. Enzymes and metabolites related to phenylpropanoid and flavonoid biosynthesis pathways are implicated in conferring higher resistance to the Cordoba population, suggesting their potential as resilience markers. Noteworthy candidates include glyoxalase I, glutathione reductase, thioredoxin reductase, and cinnamyl alcohol dehydrogenase. Overall, the manuscript is interesting; however, more extensive revisions are needed. A very important aspect is the characterisation of the isolate used for the trials. It appears to have been characterised in previous work only at morphological level.  I also reccomend separating the results section from the discussion. 

Author Response

(The authors gave the same response as above.)

Round 2

Reviewer 1 Report

Comments and Suggestions for Authors

Dear Authors,

I was asked for a second round of review for your manuscript "Proteomic and metabolomic analysis of the Quercus ilex-Phytophthora cinnamomi pathosystem reveals a population-specific response, independent of co-occurrence of drought", submitted for publication in Biomolecules.

Reading the revised version, I was pleased to see that you have successfully implemented most of my recommendations for improvement from the first round of review. Your manuscript has visibly improved, and I will recommend it for publication in Biomolecules after a final round of minor corrections that remain to be done so that your paper can make the maximum scientific impact.

I apologize if I am now making some of the remarks that stood already in the first round of review. In the first round of review there were too many things to be corrected, and some more subtle flaws may be spotted only when there are no issues with the surrounding text.

·         Abstract, line 18: please add "acorns collected from" (acorns collected from both symptomatic... and non-symptomatic...)

·         line 152-153: Drought was imposed by withholding water for 28 days, alright. But in contrast to what? Please add here the details about the watering regime for the non-drought treatments.

·         lines 345, 360, 362: here you refer to "differences between individuals" ("the two Hu individuals" - lines 345 and 360; "between individuals within the Co population" - line 362). This is confusing and needs to be rephrased. What do you mean by "the two Hu individuals"? I believe that you did not collect acorns from only two trees in Huelva. I suspect that you are using the word "individuals" incorrectly here, and that you mean something else - but I'm unable to guess what. Please rephrase.

The rest of your manuscript is excellent, I look forward to reading its published version in Biomolecules.

Kind regards,

Reviewer 1

Author Response

Please, see the attached document.

Reviewer 2 Report

Comments and Suggestions for Authors

After the revisions being done, the manuscript is recommended to be accepted. 

Author Response

Dear reviewer, thank you for submitting the second revision of the manuscript. We appreciate your efforts to enhance the quality of the document.